# The Everything-Is-a-Quantum-Wave Interpretation of Quantum Physics

**Vlatko Vedral**

Clarendon Laboratory, University of Oxford, Parks Road, Oxford OX1 3PU, UK; v.vedral@physics.ox.ac.uk

**Abstract:** In this paper, I would like to outline what I think is the most natural interpretation of quantum mechanics. By natural, I simply mean that it requires the least amount of excess baggage and that it is universal in the sense that it can be consistently applied to all the observed phenomena, including the universe as a whole. I call it the "Everything is a Quantum Wave" Interpretation (EQWI) because I think this is a more appropriate name than the Many Worlds Interpretation (MWI). The paper explains why this is so.

**Keywords:** quantum; waves; many worlds interpretation

## 1. Introduction

Let me dive straight into explaining what I have in mind. According to quantum physics, everything is actually made up of waves, but these are quantum waves (or q-waves for short), meaning that the entities that are doing the waving are what Dirac called q-numbers (as opposed to the ordinary c-numbers, "c" being classical). Mathematically, this entails having a set of (generally non-commuting) operators specified at every point in space and at every instance of time. These operators satisfy one wave equation or another, in other words they causally propagate at some finite speed (light or otherwise). This q-wave picture emerged through the work of Heisenberg [1], Jordan [2], von Neumann [3,4], Mott [5], Darwin [6,7], Schrödinger [8], Everett [9] (all standing on the shoulders of Hamilton), and many others [10–14] who have all more or less reached the same conclusion.

Let me explain a bit more how everything is a q-wave and why this presents us with the best picture of reality at present. First, there was a problem. Remember that before quantum physics, we had two fundamental entities in the world, waves and particles; however, quantum physics unified the two notions into one, leading to the well-known wave–particle dualism. However, if, according to quantum physics, particles are waves, the key phenomenon to explain in the 1920s was the observation of the alpha-particle decay in a cloud chamber. This experiment seemed to present a paradox for quantum physics.

An alpha-particle is a Helium nucleus (two protons and two neutrons) and it sometimes gets ejected in the nuclear decay of a larger nucleus. A cloud chamber was a great invention in which to observe such particles (worthy of several Nobel Prizes), though nowadays you can make one in 15 min in your own house with the usual kitchen utensils (there are many YouTube videos on this). The idea, as the name (cloud chamber) suggests, is to have a particle travel through a gas that can readily be ionized by collisions with the particle (thereby creating a cloud). As the particle collides with the gas molecules, it ionizes them in succession. Ionisation attracts neighbouring gas which condenses around the ionized molecules. Therefore, the travelling and colliding particle leaves a track of condensed vapour in its wake. To date, so good, but the problem was that the tracks are always straight lines. If, as quantum physics suggests, everything is a wave, why do we get straight lines from alpha-particles? Why not concentric circles, just like waves spreading in a pond when we throw a stone in it?

Heisenberg was the first person to explain this using their Uncertainty Principle. The emitted alpha-particle, he said, starts out as a wave, but the first molecule it hits localizes it to a small region of space (roughly the size of that molecule). In other words, this collision acts like a measurement of the position of the alpha-particle. However, the more accurately the position is determined (i.e., the more strongly the alpha particle is localized), the less determined is its momentum and, so, subsequently, the alpha-particle starts to spread as it travels, just like a wave would. However, the next collision comes pretty soon as the gas in the chamber is dense. This again focuses the particle's position reducing its position uncertainty. From there onwards, the rapid chain of collisions with the gas acts like a sequence of measuring devices that do not allow the alpha particle to spread out like a wave. It therefore leaves a straight track just as a particle would!

Mathematically, this is very simple to understand through Heisenberg's "matrix mechanics". The basic classical formula for particles motion with no forces acting on it is $x(t) = x(0) + p/mt$. Quantum mechanically, $x$ and $p$ (but not $t$ and $m$) are operators (this was Heisenberg's route to quantum physics: keep the classical dynamics, but reinterpret some quantities algebraically differently). Taking the commutator with $x(0)$ on both sides yields:

$$[x(t), x(0)] = [x(0), x(0)] + \frac{[p, x(0)]}{m}t . \tag{1}$$

Recalling that $[p, x(0)] = i\hbar$, leads us to conclude that:

$$[x(t), x(0)] = \frac{i\hbar}{m}t . \tag{2}$$

We note that the later position commutes less and less with the initial position under free evolution. This implies that

$$\Delta x(t) = \frac{\hbar}{m\Delta x(0)}t . \tag{3}$$

In other words, the trajectory of a free particle spreads out with time. In this sense, particles in quantum mechanics behave just like waves in classical physics; they diffract (and interfere). However, if the particle suddenly becomes localised through interaction with the gas, the spreading then restarts and so long as the time between the collisions is not too large (so that $\Delta x(t) \approx \Delta x(0)$), this process clearly leads to a straight trajectory. In the optical wave parlance, this is like having a laser beam of light broadening as it propagates but then encountering a sequence of slits, each of which re-focuses it and narrows it down to its original size.

## 2. The Core Argument

It is a magical explanation of how particle-like behaviour arises from waves, and it seems to make sense. However, in 1929 Mott went even further. He actually set the scene for the Many Worlds Interpretation of quantum mechanics (which, as I advocate here, should actually be called EQWI). Mott said that a single particle simultaneously traverses all the tracks in all physically allowed directions, meaning that all possible trajectories exist in a superposition and at the same time. Even though a single alpha-particle takes all the paths simultaneously and in all the directions, when we look at it, we can only see one of these trajectories. Darwin actually realised this even before Mott [15] as is clear from the following statement [6]: "so without pretending to have mastered the details, we can understand how it is possible for the $\psi$ function, so to speak, not to know in what direction the track is to be, but yet to insist that it should be a straight line. The decision as to actual track can be postponed until the wave reaches the uncovered part, where the observations are made".

It is simpler to model this in the Schrödinger picture of quantum physics, which is how Mott himself (as well as Darwin) approached the problem. The total wave-function (un-normalised) for the alpha particle and the gas is given by

$$|\Psi\rangle = \sum_n |\alpha_n\rangle |\xi_n\rangle \tag{4}$$

where $\alpha_m$ is the m-th trajectory of the alpha particle and $\xi_m$ is the state of the excited atoms of the gas along that trajectory. The fact that when one atom is excited the probability is high for the next excited atom to lie on a straight line connecting the two follows from Huygens' principle (which applies to wave-functions in quantum physics, and operators in quantum field theory, the same way that it applies to waves in classical optics—this is because the equations that quantum waves obey are the same as classical waves, it is just that—in quantum physics—the entities that obey them are q- instead of c-numbers).

However, and this is the crux of the matter, making an observation can also be described with quantum waves, so everything is unified and consistent. The reason for the fact that we only see one trajectory at a time is that even though everything is a q-wave within which things exist at the same time, when we interact with this q-wave we can only reveal some of its aspects, one at a time (this is where Heisenberg's Uncertainty comes from). We ourselves are also a collection of q-waves and it is when our q-waves correlate with the q-waves of the alpha-particle that c-numbers emerge. These correlations between q-waves are called quantum entanglement and so the classical world owes its own existence to quantum entanglement. The entangled state in Equation (4) clearly illustrates this point.

This kind of logic works at all levels and there is never any need to introduce ad hoc assumptions, such as that of a "spontaneous collapse" (which, in addition, also leads to irreversible dynamics, contrary to quantum physics). In quantum physics, even a collision between two particles is actually described as an interaction between two q-waves. This constitutes our most accurate description of nature, called quantum field theory. A particle in this theory is just one stable configuration of the underlying q-wave (or, a single excitation of the quantum field, in a more formal language of quantum field theory).

In fact, quantum field theory is the ultimate expression of the view that everything is a quantum wave. The alpha particle experiment does not need the full quantum field theory since all the particles involved are stable throughout and we need not consider their creation and annihilation. We could have completed the analysis with the full quantum field theory formalism, which would entail treating the wave-function as a field operator, but this would just have been an unnecessary overkill (actually, the whole of quantum field theory could also be performed in the Schrödinger picture, in which case the states of fields become functionals; this fact, however, does not change the logic of my argument). Mathematically, the treatment is no more complicated than solving the Schrödinger equation in the first place.

The bottom line is that reality emerges from interactions of q-waves with other q-waves. There is also no need to introduce a special classical measurement apparatus, or conscious observers or anything like that. Schrödinger, in lectures given towards the end of their life [8], clearly spelt out the same picture of quantum physics according to which everything is a q-wave. He advocated this view not only because it avoids the confusion arising from the dualistic wave–particle language (since particles are of secondary importance, being as they are specific excitations of q-waves) but also because it contains no collapses of the wave-function, no abrupt discontinuities due to measurements and no quantum jumps (as I said, the quantum wave interpretation has the least amount of excess baggage; Schrödinger was particularly keen to avoid quantum jumps, about which he said that if they turned out to be true he had wished he was a plumber and not a physicist).

Everett usually receives the credit for promoting the picture in which the whole universe is quantum and measurements are just entanglements between different quantum systems, however, as I have argued, many other physicists reached the same conclusion

well before them (as the famous cat thought experiment testifies to, Schrödinger's did so some 20 odd years before Everett). Everett emphasized the relative nature of quantum observations, meaning that relative to my state of being happy, the state of the cat is alive, while—at the same time—there is another simultaneously existing branch (you can also call it a path or a track or what-have-you) of the quantum state in which the cat is dead and I am sad. These two branches are orthogonal, but they could—at least in principle—be interfered, which is how we test their simultaneous existence. Without this interference, each branch has their own "classical" reality and one would never know that they existed in a superposition unless one was able to perform interference on them.

In the modern jargon, when one system maximally entangles to another, both systems lose coherences in their respective bases that become correlated to each other. This loss of coherence is known as decoherence. Decoherence is not another phenomenon that needs to be added to quantum physics in order to explain the emergence of classicality. It is already contained within quantum physics and emerges naturally whenever there is interaction.

So why EQWI instead of MWI? Precisely because the state of the universe where we can talk about the worlds is just a limiting, special case of EWQI. The worlds only emerge fully when we have fully orthogonal states of observers (i.e., the quantum systems performing measurements, and measurements are—in this interpretation—just entangling unitaries with other systems). Otherwise the classical reality is only approximate. To a high degree of accuracy, each alpha particle tract is orthogonal to every other one, which means that you can think of them as different worlds. This state is analogous to the Fraunhoffer, or far-field limit in wave optics. At the other end is the Fresnel, or near-field limit, and it would correspond to the quantum state that does not allow us to talk about the separate worlds since different branches have a high degree of overlap. However, we know that they exist at the same time because all these paths could—at least in principle—be brought together to interfere. The possibility of being able to interfere different worlds is crucial to this view and leads to the fact that the "unobserved outcomes can affect future measurements" as Deutsch's version of Schrödinger's cat experiment has taught us. I have written about this elsewhere [16–18], and recommend it to the interested reader (see also [19,20]).

This way of thinking about quantum physics, namely that everything is a quantum wave, a quantum field whose relevant q-numbers can be specified at every point in space and at every instance of time, automatically inherits one important feature of classical field theory. Quantum fields too (just like classical fields) can be constructed so as not to allow action at a distance, i.e., using fields enables us to keep the principle that all interactions are local in space (i.e., no interaction takes place instantaneously at a distance). In this sense, the EQWI is as local as Maxwell's electrodynamics, even though the elements of reality in quantum physics, the q-numbers, are very different from their classical counterparts.

It is sometimes said that this quantum wave-like view of reality is incapable of explaining the origin of probabilities since everything is always seen and phrased only at the level of amplitudes. However, this is not true and both the single-shot notion of probability (such as in the notion of the "degree of belief") as well as the frequentist one (such as is obtain in a ensemble of identically prepared quantum systems) can be derived from the quantum waves. The point being that different probabilities are emergent, derived notions when one takes the quantum waves as the primary entities. In this sense, even a single "click" in a photodetector is an extraordinarily complex phenomenon if one wants to reduce it to the interactions between quantum fields (which can be performed, though, in practice, there is hardly even a reason to do so). The current exposition is clearly not the place to go into these details and the interested reader is referred to [21] and references therein.

## 3. Conclusions

Finally, the everything is a q-wave interpretation is uniquely quantum, but I would like to conclude by explaining how its existence owes everything to Hamilton's version of classical physics. Hamilton died well before the birth of quantum physics, so how could he

have anticipated all this? This is because Hamilton discovered an ingenious way of doing Newtonian physics that ultimately paved the way to quantum physics.

Hamilton thought of particles moving in straight lines as rays of light moving in a uniform medium. When a force acted on a particle to change its direction of motion, this was for Hamilton analogous to light entering a denser medium and refracting (bending). Hamiltonian mechanics therefore uses the methodology of waves (things like wavefronts, rays, refractive indices, etc.) to describe the mechanics of particles (things like trajectories, forces, accelerations, etc.). If we could resurrect him, Hamilton would have no problem understanding the alpha-particle tracks in a cloud chamber. In fact the relationship between their wave and particle mechanics is the classical analogue of the relationship between EQWI and MWI. The "only" thing he was missing at the time were the q-numbers. There simply was no need for them (i.e., no experimental evidence, such as a particle in a superposition of different locations, to force us to use them) prior to the twentieth century. Otherwise, Hamilton would have probably written down the Schrödinger equation some fifty years before Schrödinger. All of the challenges that we are facing when trying to understand quantum physics are related to the fact that the fundamental entities are q-numbers, and that, unlike the c-numbers, they do not correspond to individual measurement outcomes. The classical world of c-numbers is a consequence of quantum entanglement.

It is also in the spirit of Hamilton that we can phrase quantum dynamics in a timeless way. Namely, we could just use the wave-function of the universe written in 3-space and think of different elements in this superposition as different times [22]. One might call this picture of "different universes being different times" the ultimate expression of the EQWI. However, no matter how we choose to represent the evolution of quantum waves, the interpretation advocated here remains valid for all of them.

I hope I have convinced you that this is the most natural picture of the universe we have at present. I don't for one second believe that it is our final picture. What lies beyond is, of course, wide open, and we have to wait for the next theory of physics to be able to talk about its interpretation. The next theory of physics will have to contain quantum physics as a special limiting case which means that the q-waves are here to stay with us and whatever notion extends and replaces them in the new theory will be at least as weird, but more likely much weirder than the operators we have at present.

**Funding:** This research received no external funding.

**Data Availability Statement:** Not Applicable.

**Acknowledgments:** V.V. is grateful to the Moore Foundation for supporting their research.

**Conflicts of Interest:** The author declares no conflict of interest.

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
