# Peer review of "The Everything-Is-a-Quantum-Wave Interpretation of Quantum Physics"

_quantumrep, doi:10.3390/quantum5020031_

Round 1
Reviewer 1 Report
I am sorry to say, but this article rather reads as written for any kind of popular media than for a serious scientific journal. I do not recommend further consideration for publication.
Reviewer 2 Report
Review for quantumrep-2274654
This is interesting, but I worry that it doesn't really address the questions which interpretations of quantum mechanics -- like EQWI -- ought to address. It reads more like a rephrasing of things that have already, quite often, been said by physicists and philosophers -- and at that, a rephrasing which glosses over the difficult and interesting issues which really should be engaged with. In what follows, I give a few concrete examples of this glossing over.
Major comments
``[T]he better we know the position, the less we know the momentum and, so, subsequently, the alpha-particle starts to spread as it travels just like a wave'' (p. 1).
This makes it sound like the alpha-particle starts to spread out because of what we know. And for several reasons, that seems wrong: our knowledge of the alpha-particle does not seem capable of influencing it's motion. First, imagine a baby who knows nothing about quantum mechanics looking at the cloud chamber. The baby's knowledge is different from the knowledge of a physicist who could also look at the chamber. But the alpha-particle, watched by the baby, propagates in the same way as the alpha-particle watched by the physicist. Second, Heisenberg's whole explanation of this particular phenomenon takes the measurements of the alpha-particle to be given, somehow or other, by its collisions with molecules in the cloud chamber...so even assuming that's right -- which I doubt it is -- then its very weird to appeal to our relative knowledge of the position/momentum of the particle...surely it's the knowledge of the molecules in the chamber to which Heisenberg should've appealed, since those are the things doing the measuring on his view? (and of course that appeal wouldn't make sense either...molecules in chambers don't have mental states like knowledge)
The author claims that c-numbers emerge when out q-waves correlate with the q-waves of the alpha-particle. And the author claims that these correlations between q-waves -- quantum entanglement -- to which the classical world owes its existence (p. 2). And the author claims that ``[t]his kind of logic works at all levels and there is never any need to introduce ad hoc assumptions such as that of a `spontaneous collapse' '' (p. 2).
None of this, however, seems to answer the main questions, related to collapse and the emergence of a quasi-classical world, which interpretations of quantum theory should answer. This just seems like a -- very standard -- rephrasing of the issues at hand. And in the terminology of this (very standard) rephrasing, one of the main questions is this: what is this `owing' relation that the author mentions, when claiming that the classical world `owes its existence' to correlations between q-waves? What principles, exactly, describe the way that those correlations generate the classical world? Simply saying that the classical world `owes its existence' to this isn't really a theory of where the classical world comes from.
Similarly, later on, the author writes that ``reality emerges from interactions of q-waves with other q-waves'' (p. 2). How? I'm actually pretty sympathetic to the view that the author is proposing here: I agree, actually, that reality is somehow or other generated by interactions among quantum-theoretic waves. But the interesting question, in this arena, isn't whether or not reality does emerge from those interactions: obviously it does, since (i) reality exists, and (ii) those interactions exist, and (iii) physical theories use those interactions to explain macro-phenomena like pointer positions and tracks in cloud chambers and so on. The interesting question, rather, is: exactly how does reality emerge from those interactions. Again: what are the precise principles describing that emergence?
Minor comments
1. `could chamber' should be `cloud chamber' (p. 1).
2. ``The worlds [of MWI] only emerge when we have fully orthogonal states of observers'' (p. 3). This seems like an overly strong requriement on emergence. For lots of decoherence-related reasons, most proponents of MWI think that the worlds can emerge when we only have approximately (not fully) orthogonal states of a certain sort (which may or may not be states of observers).
Reviewer 3 Report
This paper presents an interesting "Everything-is-a-Quantum-Wave" interpretation of quantum theory. According to this interpretation, everything is a quantum wave, a quantum field whose relevant q-numbers can be specified at every point in space and at every instance of time. This provides an ontology in 3-space for quantum theory. I would like to see how this ontology extends to entangled states in 3-space, and how the interpretation solves the problem of probability of MWI. I recommend publication with minor revision.
Reviewer 4 Report
Vlatko Vedral is a well-known and highly cited theoretical physicist, but unfortunately what he writes in this paper makes very little sense to me. Let us start from his sentence ‘According to quantum physics, everything is actually made up of waves, but these are quantum waves (or q-waves for short), meaning that the entities that are doing the waving are what Dirac called q–numbers (as opposed to the ordinary c-numbers, “c” being classical)’.
I have been using and doing quantum physics for more than 40 years, but I see no evidence whatsoever that ‘everything is actually made up of waves’. From obvious empirical evidence we can get that (i) the physical world is made of objects, properties, and events, and (ii) as far as quantum physics is concerned, probabilities must be used to describe these objects, properties, and events, and classical probabilities cannot do the job, as shown by a huge list of no-go theorems. So a non-classical probability theory must be developed, and this is what the basic quantum formalism is about. Certainly this formalism allows for interference effects, and wavy behaviors, that are quite unusual for probabilities, this is just why nonclassical probabilities are required. But concluding that ‘everything is actually made up of waves’ sounds as meaningless to me as if a statistician would pretend that all the populations he is studying are actually made of normal distributions.
So clearly I have a major disagreement with the author about the meaning of ‘everything is actually made up of…’, and in his statements I see a confusion between physical and mathematical objects. For me the mathematical objects or concepts do not define physical reality, they describe it, as a (sophisticated) language can do. Clearly this is not the point of view of this article, where the mathematical solution of Schrödinger’s equation ‘is what it is’. So my disagreement is of deep philosophical nature, and what is written in this paper simply does not make sense to me.
However, the article is submitted to a special issue on MWI, and it is known that the philosophical position of MWI supporters is much closer to the one presented in this article, than to mine: they want to save determinism and ‘universal unitarity’, at any price. Given this context, I have no objection to publication of this article in the special issue, as one more possible MWI/EQWI option.
Round 2
Reviewer 1 Report
My previous report holds true. In his answer to the refs, the author does not address my concerns.
Reviewer 2 Report
I am happy to accept this in its present form.